# Collaborative Deterministic–Probabilistic Forecasting for Diverse Spatiotemporal Systems

## Abstract

Probabilistic forecasting is crucial for real-world spatiotemporal systems, such as climate, energy, and urban environments, where quantifying uncertainty is essential for informed, risk-aware decision-making. While diffusion models have shown promise in capturing complex data distributions, their application to spatiotemporal forecasting remains limited due to complex spatiotemporal dynamics and high computational demands. we propose **CoST**, a general forecasting framework that **Co**llaborates deterministic and diffusion models for diverse **S**patio**T**emporal systems. CoST formulates a mean-residual decomposition strategy: it leverages a powerful deterministic model to capture the conditional mean and a lightweight diffusion model to learn residual uncertainties. This collaborative formulation simplifies learning objectives, improves accuracy and efficiency, and generalizes across diverse spatiotemporal systems. To address spatial heterogeneity, we further design a scale-aware diffusion mechanism to guide the diffusion process. Extensive experiments across ten real-world datasets from climate, energy, communication, and urban systems show that CoST achieves 25% performance gains over state-of-the-art baselines, while significantly reducing computational cost. Code and datasets are available at: `https://anonymous.4open.science/r/CoST_17116`.

## 1 Introduction

Real-world spatiotemporal systems underpin many critical domains, such as climate science, energy systems, communication networks, and urban environments. Accurate forecasting of the dynamics is essential for planning, resource allocation, and risk management (Xie et al., 2020; Boussif et al., 2023; Xu et al., 2021; Sheng et al., 2025). Existing approaches fall into two categories: deterministic and probabilistic forecasting. Deterministic methods estimate the conditional mean by minimizing MAE or MSE losses to capture spatiotemporal patterns (Zhang et al., 2017; Ma et al., 2024; Yuan et al., 2024). In contrast, probabilistic methods aim to learn the full predictive distribution of observed data (Rasul et al., 2021; Li et al., 2024; Yuan & Qiao, 2024), enabling uncertainty quantification to support forecasting. This is particularly important in many domains, for example, in climate modeling and renewable energy, where assessing prediction reliability is essential for risk-aware decisions such as disaster preparedness and energy grid management (Palmer, 2012; Vargas Zeppetello et al., 2022).

In this paper, we highlight the critical role of probabilistic forecasting in capturing uncertainty and improving the reliability of spatiotemporal predictions. However, it is non-trivial due to three challenges. First, these systems exhibit complex evolving dynamics, characterized by periodic trends, seasonal variations, and stochastic fluctuations (Cao et al., 2021; Yuan & Qiao, 2024). Second, these systems involve intricate spatiotemporal interactions and nonlinear dependencies (Jiang et al., 2017; Yuan et al., 2024). Third, real-world applications require both computationally efficient and scalable models (Palmer, 2014; Sheng et al., 2025). Recently, diffusion models have been widely adopted for probabilistic forecasting (Wen et al., 2023; Yuan & Qiao, 2024; Rasul et al., 2021; Sheng et al., 2025). Compared with existing approaches such as Generative Adversarial Networks (GANs) (Goodfellow et al., 2020; Gao et al., 2022) and Variational Autoencoders (VAEs) (Kingma, 2013; Li et al., 2022), diffusion models offer superior capability in capturing complex data distributions while ensuring stable training (Ho et al., 2020; Tashiro et al., 2021; Ho et al., 2022). These advantages make diffusion models a promising alternative. However, originally developed for image generation, they

face inherent limitations in capturing temporal correlations in sequential data, as evidenced in video generation (Zhang et al., 2024; Qiu et al., 2023; Chang et al., 2024; Ge et al., 2023) and time series forecasting (Yuan & Qiao, 2024; Rühling Cachay et al., 2023; Shen & Kwok, 2023).

To address this issue, recent efforts have explored incorporating temporal correlations as conditional inputs to guide the diffusion process (Rasul et al., 2021; Shen & Kwok, 2023; Wen et al., 2023), or injecting temporal priors into the noised data to explicitly model temporal correlations across time steps (Li et al., 2024; Sheng et al., 2025; Yuan & Qiao, 2024). While these approaches improve temporal modeling, they remain constrained by the inherent limitations of the diffusion framework (Li et al., 2024; Rühling Cachay et al., 2023). In contrast, we introduce a new perspective: rather than relying solely on diffusion models to capture the full

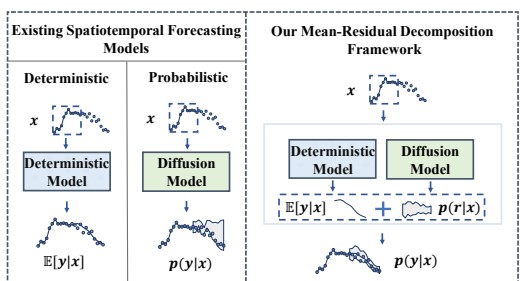

Figure 1: Comparison of existing models with our mean-residual decomposition framework.

data distribution, we propose a collaborative approach that combines a deterministic model and a diffusion model, leveraging their complementary strengths for probabilistic forecasting. Our design offers two key advantages. First, by leveraging powerful deterministic models to predict the conditional mean, it effectively captures the primary spatiotemporal patterns and benefits from advancements in established architectures. Second, instead of requiring the diffusion model to learn the full data distribution from scratch, we employ it to model the residuals, focusing its capacity on capturing uncertainty beyond the mean. This collaborative framework simplifies the learning objectives for each component and enhances both predictive accuracy and probabilistic expressiveness.

Building on this insight, we propose **CoST**, a general forecasting framework that **Co**llaborates deterministic and diffusion models for a wide range of **S**patio**T**emporal systems. As illustrated in Figure 1, we first leverage an advanced deterministic spatiotemporal forecasting model to estimate the conditional mean $\mathbb{E}[y|x]$, effectively capturing the regular patterns. Based on this, we model the residual distribution $p(r|x) = p((y - \mathbb{E}[y|x])|x)$ using a diffusion model, which complements the deterministic forecasting with uncertainty quantification. Since the diffusion model focuses solely on residuals, it allows us to adopt a lightweight denoising network and mitigate the computational overhead associated with multi-step diffusion processes. To address spatial heterogeneity, we quantify differences across spatial units and introduce a scale-aware diffusion mechanism. More importantly, we propose a comprehensive evaluation protocol for spatiotemporal probabilistic forecasting by incorporating metrics such as QICE and IS, rather than relying solely on traditional measures like CRPS, MAE, and RMSE. In summary, our main contributions are as follows:

- We highlight the importance of probabilistic forecasting for complex spatiotemporal systems and introduce a novel perspective that integrates deterministic and probabilistic modeling in a collaborative framework.

- We propose **CoST**, a mean-residual decomposition approach that employs a deterministic model to estimate the conditional mean and a diffusion model to capture the residual distribution. We further design a scale-aware diffusion mechanism to address spatial heterogeneity. CoST is broadly applicable across a wide range of critical real-world spatiotemporal systems.

- Extensive experiments on ten real-world datasets spanning climate science, energy systems, communication networks, and urban environments show that CoST consistently outperforms state-of-the-art baselines on both deterministic and probabilistic metrics, achieving an average improvement of 25% while offering notable gains in computational efficiency.

## 2 RELATED WORK

We have provided definitions and related work on spatiotemporal deterministic forecasting and probabilistic forecasting in Appendix B.

**Diffusion-based spatiotemporal probabilistic forecasting.** Most diffusion-based forecasting methods formulate the task as conditional generation without explicitly modeling temporal dynamics,

which hinders the generation of temporally coherent sequences (Tashiro et al., 2021; Gao et al., 2023; Wen et al., 2023; Rühling Cachay et al., 2023). Moreover, the progressive corruption of time series during diffusion often distorts key patterns like long-term trends and periodicity, making temporal recovery difficult (Yuan & Qiao, 2024; Liu et al., 2023). To address this, methods such as TimeGrad (Rasul et al., 2021) and TimeDiff (Shen & Kwok, 2023) incorporate temporal embeddings as conditional inputs to enhance temporal awareness. Other approaches like NPDiff (Sheng et al., 2025) and Diffusion-TS (Yuan & Qiao, 2024) inject temporal priors into the diffusion process to better preserve temporal dynamics. More recently, DYffusion (Rühling Cachay et al., 2023) redefines the denoising process to explicitly model temporal transitions at each diffusion step. Unlike prior methods, we avoid using diffusion to model temporal dynamics. Instead, we decouple forecasting into deterministic mean prediction and residual uncertainty estimation, allowing the diffusion model to focus exclusively on modeling the residual distribution.

**Hybrid Deterministic and Diffusion-based Forecasting Models.** A promising recent trend involves hybridizing deterministic and diffusion models to achieve better results (Mardani et al., 2023; Gong et al., 2024; Li et al., 2024; Yu et al., 2024). For instance, CorrDiff (Mardani et al., 2023) refines the output of a deterministic UNet using a diffusion model for high-resolution atmospheric downscaling. Similarly, CasCast (Gong et al., 2024) adopts a cascaded pipeline where the coarse global prediction from a deterministic model guides a latent diffusion model to generate a refined full future state. In time series forecasting, TMDM (Li et al., 2024) conditions its diffusion process on the output of a transformer model to better learn the full data distribution. DiffCast (Yu et al., 2024) takes a coupled, end-to-end training approach for precipitation nowcasting, jointly optimizing a deterministic backbone with a diffusion model. In contrast, our approach, CoST, introduces a two-stage mean-residual decomposition strategy that distinctly separates mean prediction and residual distribution modeling. More importantly, while prior methods are typically tailored to a single domain, CoST is designed as a general framework that incorporates spatiotemporal considerations and residual heterogeneity, making it broadly applicable across diverse spatiotemporal systems.

# 3 PRELIMINARIES

We provide a summary of notations used in this paper in Appendix C.1 for clarity.

**Spatiotemporal systems.** Spatiotemporal systems underpin many domains such as climate science, energy, communication networks, and urban environments. The data recording spatiotemporal dynamics are typically represented as a tensor $\mathbf{x} \in \mathbb{R}^{T \times V \times C}$, where $T$, $V$, and $C$ denote the temporal, spatial, and feature dimensions, respectively. Depending on the spatial structure, the data can be organized as grid-structured ($V = H \times W$) or graph-structured (where $V$ represents the set of nodes). Given a historical context $\mathbf{x}^{co} = \mathbf{x}^{t-M+1:t}$ of length $M$, the goal is to predict future targets $\mathbf{x}^{ta} = \mathbf{x}^{t+1:t+P}$ over a horizon $P$ using a model $\mathcal{F}$.

**Conditional diffusion models.** The diffusion-based forecasting includes a forward process and a reverse process. In the forward process, noise is added incrementally to the target data $\mathbf{x}_0^{ta}$, gradually transforming the data distribution into a standard Gaussian distribution $\mathcal{N}(\mathbf{0}, \mathbf{I})$. At any diffusion step, the corrupted target data can be computed using the one-step forward equation:

$$\mathbf{x}_n^{ta} = \sqrt{\bar{\alpha}_n}\mathbf{x}_0^{ta} + \sqrt{1 - \bar{\alpha}_n}\epsilon, \quad \epsilon \sim \mathcal{N}(\mathbf{0}, \mathbf{I}), \tag{1}$$

where $\bar{\alpha}_n = \prod_{i=1}^{n} \alpha_i$ and $\alpha_n = 1 - \beta_n$. In the reverse process, prediction begins by first sampling $\mathbf{x}_N^{ta}$ from the standard Gaussian distribution $\mathcal{N}(\mathbf{0}, \mathbf{I})$, followed by a denoising procedure through the following Markov process:

$$p_\theta(\mathbf{x}_{0:N}^{ta}) := p(\mathbf{x}_N^{ta}) \prod_{n=1}^{N} p_\theta(\mathbf{x}_{n-1}^{ta}|\mathbf{x}_n^{ta}, \mathbf{x}_0^{co}),$$

$$p_\theta(\mathbf{x}_{n-1}^{ta}|\mathbf{x}_n^{ta}) := \mathcal{N}(\mathbf{x}_{n-1}^{ta}; \mu_\theta(\mathbf{x}_n^{ta}, n|\mathbf{x}_0^{co}), \Sigma_\theta(\mathbf{x}_n^{ta}, n)), \tag{2}$$

$$\mu_\theta(\mathbf{x}_n^{ta}, n|\mathbf{x}_0^{co}) = \frac{1}{\sqrt{\bar{\alpha}_n}} \left( \mathbf{x}_n^{ta} - \frac{\beta_n}{\sqrt{1 - \bar{\alpha}_n}}\epsilon_\theta(\mathbf{x}_n^{ta}, n|\mathbf{x}_0^{co}) \right)$$

where the variance $\Sigma_\theta(\mathbf{x}_n^{ta}, n) = \frac{1 - \bar{\alpha}_{n-1}}{1 - \bar{\alpha}_n}\beta_n$, and $\epsilon_\theta(\mathbf{x}_n^{ta}, n|\mathbf{x}_0^{co})$ is predicted by the denoising network trained by the loss function below:

$$\mathcal{L}(\theta) = \mathbb{E}_{n, \mathbf{x}_0, \epsilon} \left[ \left\| \epsilon - \epsilon_\theta(\mathbf{x}_n^{ta}, n|\mathbf{x}_0^{co}) \right\|_2^2 \right]. \tag{3}$$

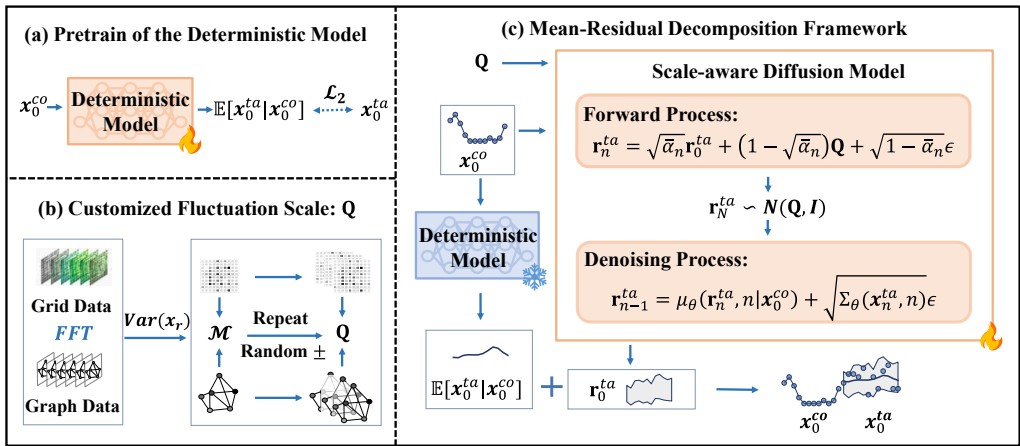

Figure 2: Overview of CoST: (a) Pretraining of the deterministic model; (b) Computation of the customized fluctuation scale; (c) Overall framework of the mean-residual decomposition.

**Evaluations of probabilistic forecasting.** We argue that probabilistic forecasting should be assessed from two key perspectives: *Data Distribution*—the predicted distribution should match the empirical distribution, and *Prediction Usability*—prediction intervals should achieve high coverage while remaining sharp. While metrics like CRPS, MAE and RMSE are widely used, they fail to assess: **(i)** the accuracy of quantile-wise coverage; **(ii)** whether the interval width reflects true uncertainty. To address this, we introduce Quantile Interval Coverage Error (QICE) (Han et al., 2022) and Interval Score (IS) (Gneiting & Raftery, 2007) as complementary metrics.

*(i) QICE* measures the mean absolute deviation between the empirical and expected proportions of ground-truth values falling into each of equal-sized quantile intervals. QICE evaluates how well the predicted distribution aligns with the expected coverage across quantiles, which is defined as follows:

$$\text{QICE} := \frac{1}{M_{\text{QIs}}} \sum_{m=1}^{M_{\text{QIs}}} \left| r_m - \frac{1}{M_{\text{QIs}}} \right|, \quad r_m = \frac{1}{N} \sum_{n=1}^{N} \mathbb{1}_{y_n \geq \hat{y}_n^{\text{low}_m}} \cdot \mathbb{1}_{y_n \leq \hat{y}_n^{\text{high}_m}}, \tag{4}$$

where $\hat{y}_n^{\text{low}_m}$ and $\hat{y}_n^{\text{high}_m}$ denote the bounds of the $m$-th quantile interval for $y_n$. Ideally, each QI should contain $1/M_{\text{QIs}}$ of the observations, yielding a QICE of 0. Lower QICE indicates better alignment between predicted and true distributions.

*(ii) IS* evaluates prediction interval (PI) quality by jointly accounting for sharpness and empirical coverage, and is defined as:

$$\text{IS} := \frac{1}{N} \sum_{n=1}^{N} \left[ (u_n^{\alpha_{CI}} - l_n^{\alpha_{CI}}) + \frac{2}{\alpha_{CI}} (l_n^{\alpha_{CI}} - y_n) \mathbb{1}_{y_n < l_n^{\alpha_{CI}}} + \frac{2}{\alpha_{CI}} (y_n - u_n^{\alpha_{CI}}) \mathbb{1}_{y_n > u_n^{\alpha_{CI}}} \right], \tag{5}$$

where $u_n^{\alpha_{CI}}$ and $l_n^{\alpha_{CI}}$ are the upper and lower bounds of the central prediction interval for the $n$-th data point, derived from the corresponding predictive quantiles. A narrower interval improves the score, while missed coverage incurs a penalty scaled by $\alpha_{CI}$. Lower IS indicates better performance.

## 4 METHODOLOGY

In this section, we propose CoST, a unified framework that combines the strengths of deterministic and diffusion models. Specifically, we first train a deterministic model to predict the conditional mean, capturing the regular spatiotemporal patterns. Then, guided by a customized fluctuation scale, we employ a scale-aware diffusion model to learn the residual distribution, enabling fine-grained uncertainty modeling. An overview of the CoST architecture is shown in Figure 2.

### 4.1 THEORETICAL ANALYSIS OF MEAN-RESIDUAL DECOMPOSITION

Current diffusion-based probabilistic forecasting approaches typically employ a single diffusion model to capture the full distribution of data, incorporating both the regular spatiotemporal patterns and the

random fluctuations. However, jointly modeling these components remains challenging (Yuan & Qiao, 2024). Inspired by Mardani et al. (2023) and the Reynolds decomposition in fluid dynamics (Pope, 2001), we propose to divide probabilistic forecasting into two parts: predicting the conditional mean and modeling the residual distribution. The spatiotemporal data $\mathbf{x}^{ta}$ can therefore be expressed as:

$$\mathbf{x}^{ta} = \underbrace{\mathbb{E}[\mathbf{x}^{ta}|\mathbf{x}^{co}]}_{:=\boldsymbol{\mu}(Deterministic)} + \underbrace{(\mathbf{x}^{ta} - \mathbb{E}[\mathbf{x}^{ta}|\mathbf{x}^{co}])}_{:=\mathbf{r}(Diffusion)}, \tag{6}$$

where $\boldsymbol{\mu}$ is the conditional mean representing the regular patterns, and $\mathbf{r}$ is the residual representing the random variations. If the deterministic model approximates the conditional mean accurately, the expected residual becomes negligible, i.e. $\mathbb{E}[\mathbf{r}|\mathbf{x}^{co}] \approx 0$, and we can obtain that $\mathrm{var}(\mathbf{r}|\mathbf{x}^{co}) = \mathrm{var}(\mathbf{x}^{ta}|\mathbf{x}^{co})$. Based on the law of total variance (Bertsekas & Tsitsiklis, 2008), we can express the variance of the target data and residuals as:

$$\mathrm{var}(\mathbf{r}) = \mathbb{E}[\mathrm{var}(\mathbf{r}|\mathbf{x}^{co})] + \underbrace{\mathrm{var}(\mathbb{E}[\mathbf{r}|\mathbf{x}^{co}])}_{=0}, \quad \mathrm{var}(\mathbf{x}^{ta}) = \mathbb{E}[\mathrm{var}(\mathbf{x}^{ta}|\mathbf{x}^{co})] + \underbrace{\mathrm{var}(\mathbb{E}[\mathbf{x}^{ta}|\mathbf{x}^{co}])}_{\geq 0}. \tag{7}$$

Due to $\mathrm{var}(\mathbf{r}|\mathbf{x}^{co}) = \mathrm{var}(\mathbf{x}^{ta}|\mathbf{x}^{co})$, we have $\mathrm{var}(\mathbf{r}) \leq \mathrm{var}(\mathbf{x}^{ta})$. Moreover, the highly dynamic nature of the spatiotemporal system results in a larger $\mathrm{var}(\mathbb{E}[\mathbf{x}^{ta}|\mathbf{x}^{co}])$, which consequently makes $\mathrm{var}(\mathbf{r})$ smaller compared to $\mathrm{var}(\mathbf{x}^{ta})$. Our core idea is that if a deterministic model can accurately predict the conditional mean, that is, $\boldsymbol{\mu} \approx \mathbb{E}_\theta[\mathbf{x}^{ta}|\mathbf{x}]$, then the diffusion model can be dedicated solely to learning the simpler residual distribution. This design avoids the challenge diffusion models face in modeling complex spatiotemporal dynamics, while fully exploiting their strength in uncertainty estimation. By collaborating high-performing deterministic architectures and diffusion models, our method effectively captures regular dynamics and models uncertainty via residual learning.

## 4.2 MEAN PREDICTION VIA DETERMINISTIC MODEL

To capture the conditional mean $\mathbb{E}_\theta[\mathbf{x}^{ta}|\mathbf{x}^{co}]$, our framework leverages existing high-performance deterministic architectures, which are designed to capture complex spatiotemporal dynamics efficiently. In our main experiments, we use the STID (Shao et al., 2022) model as the backbone for mean prediction, and also validate our framework with ConvLSTM (Shi et al., 2015), STNorm (Deng et al., 2021), and iTransformer (Liu et al.) to ensure its generality ( See Section 5.1). In the first stage of training, we pretrain the deterministic model for 50 epochs using historical conditional inputs $\mathbf{x}^{co}$ to output the mean estimate $\mathbb{E}_\theta[\mathbf{x}^{ta}|\mathbf{x}^{co}]$. The model is trained with the standard $\mathcal{L}_2$ loss:

$$\mathcal{L}_2 = \left\| \mathbb{E}_\theta[\mathbf{x}^{ta}|\mathbf{x}^{co}] - \mathbf{x}^{ta} \right\|_2^2. \tag{8}$$

## 4.3 RESIDUAL LEARNING VIA DIFFUSION MODEL

The residual distribution of spatiotemporal data is not independently and identically distributed (i.i.d.) nor does it follow a fixed distribution, such as $\mathcal{N}(0, \sigma)$. Instead, it often exhibits complex spatiotemporal dependence and heterogeneity. We use the diffusion model to focus on learning the distribution of residual $\mathbf{r}^{ta} = \mathbf{x}^{ta} - \mathbb{E}_\theta[\mathbf{x}^{ta}|\mathbf{x}^{co}]$. Accordingly, the target data $\mathbf{x}^{ta}$ for diffusion models in Eqs. equation 1, equation 2, and equation 3 is replaced by $\mathbf{r}^{ta}$. We incorporate timestamp information as a condition in the denoising process and concatenate the context data $\mathbf{x}_0^{co}$ with noised residual $\mathbf{r}_n^{ta}$ as input to capture real-time fluctuations. Notably, no noise is added to $\mathbf{x}_0^{co}$ during diffusion training or inference. To model the spatial patterns of the residuals, we propose a scale-aware diffusion process to further distinguish the heterogeneity for different spatial units. In this section, we detail the calculation of $\mathbf{Q}$ and how it is integrated into the scale-aware diffusion process.

**(i) Customized fluctuation scale.** Specifically, we apply the Fast Fourier Transform (FFT) to spatiotemporal sequences in the training set to quantify fluctuation levels in different spatial units and use the custom scale $\mathbf{Q}$ as input to account for spatial heterogeneity in residual. Specifically, we first employ FFT to extract the fluctuation components for each spatial unit within the training set. The detailed steps are as follows:

$$\mathbf{A}_k = |\text{FFT}(\mathbf{x})_k|, \quad \phi_k = \phi(\text{FFT}(\mathbf{x})_k), \quad \mathbf{A}_{\max} = \max_{k \in \left\{1, \cdots, \left\lfloor \frac{L}{2} \right\rfloor + 1\right\}} \mathbf{A}_k,$$

$$\mathcal{K} = \left\{ k \in \left\{1, \cdots, \left\lfloor \frac{L}{2} \right\rfloor + 1\right\} : \mathbf{A}_k < 0.1 \times \mathbf{A}_{\max} \right\}, \tag{9}$$

$$\mathbf{x}_{\mathbf{r}}[i] = \sum_{k \in \mathcal{K}} \mathbf{A}_k \left[ \cos\left(2\pi \mathbf{f}_k i + \phi_k\right) + \cos\left(2\pi \bar{\mathbf{f}}_k i + \bar{\phi}_k\right) \right],$$

where $\mathbf{A}_k, \phi_k$ reprent the amplitude and phase of the $k-$th frequency component. $L$ is the temporal length of the training set. $\mathbf{A}_{\max}$ is the maximum amplitude among the components, obtained using the $\max$ operator. $\mathcal{K}$ represents the set of indices for the selected residual components. $\mathbf{f}_k$ is the frequency of the k-th component. $\bar{\mathbf{f}}_k, \bar{\phi}_k$ represent the conjugate components. $\mathbf{x}_{\mathbf{r}}$ ref to the extracted residual component of the training set. We then compute the variance $\sigma_v^2$ of the residual sequence for each location $v$ and expand it to match the shape as $\mathbf{r}_0^{ta} \in \mathbb{R}^{B \times V \times P}$, where $B$ represents the batch size. And we can get the variance tensor $\mathcal{M}$:

$$\mathcal{M}_{b,v,p} = \sigma_v^2, \forall b \in \{1, \cdots, B\}, \forall v \in \{1, \cdots, V\}, \forall p \in \{1, \cdots, P\}. \tag{10}$$

The residual fluctuations are bidirectional, encompassing both positive and negative variations, so we generate a random sign tensor $\mathbf{S} \in \mathbb{R}^{B \times V \times P}$ for $\mathcal{M}$, where each element $S_{b,v,p}$ of $\mathbf{S}$ is sampled from a Bernoulli distribution with $p = 0.5$. The customized fluctuation scale $\mathbf{Q}$ is computed as:

$$\mathbf{Q}_{b,v,p} = S_{b,v,p} \times \mathcal{M}_{b,v,p}, \forall b \in \{1, \cdots, B\}, \forall v \in \{1, \cdots, V\}, \forall p \in \{1, \cdots, P\}. \tag{11}$$

Then $\mathbf{Q}$ is used as the input of the denoising network.

**(ii) Scale-aware diffusion process.** The vanilla diffusion models assume a shared prior distribution $\mathcal{N}(0, I)$ across all spatial locations, failing to capture spatial heterogeneity. To further model such differences, we adopt the technique proposed by Han et al. (2022) to make the residual learning location-specific conditioned on $\mathbf{Q}$. Specifically, we redefine the noise distribution at the endpoint of the diffusion process as follows:

$$p(\mathbf{r}_N^{ta}) = \mathcal{N}(\mathbf{Q}, I), \tag{12}$$

Accordingly, the Eq (1) in the forward process is rewritten as:

$$\mathbf{r}_n^{ta} = \sqrt{\bar{\alpha}_n} \mathbf{r}_0^{ta} + (1 - \sqrt{\bar{\alpha}_n})\mathbf{Q} + \sqrt{1 - \bar{\alpha}_n}\epsilon, \quad \epsilon \sim \mathcal{N}(0, I). \tag{13}$$

And in the denoising process, we sample $\mathbf{r}_N^{ta}$ from $\mathcal{N}(\mathbf{Q}, I)$, and denoise it use Eq (2), the computation of $\mu_\theta(\mathbf{r}_n^{ta}, n|\mathbf{x}_0^{co})$ in Eq (2) is modified as:

$$\mu_\theta(\mathbf{r}_n^{ta}, n|\mathbf{x}_0^{co}) = \frac{1}{\sqrt{\bar{\alpha}_n}}\left(\mathbf{r}_n^{ta} - \frac{\beta_n}{\sqrt{1 - \bar{\alpha}_n}}\epsilon_\theta(\mathbf{r}_n^{ta}, n|\mathbf{x}_0^{co})\right) + (1 - \frac{1}{\sqrt{\bar{\alpha}_n}})\mathbf{Q}. \tag{14}$$

This modification allows the diffusion process to be conditioned on location-specific priors $\mathbf{Q}$, enhancing its ability to model spatial heterogeneity in uncertainty.

## 4.4 TRAINING AND INFERENCE

We adopt a two-stage training procedure: first pretraining a deterministic model to predict the conditional mean, then training a diffusion model to capture the residual distribution (Algorithm 1). During inference, the deterministic model provides the mean prediction, while the diffusion model estimates residuals; their outputs are combined to form the final forecast (Algorithm 2).

## 5 EXPERIMENTS

**Datasets.** We evaluate our method on ten datasets spanning four domains, including climate (SST-CESM2 and SST-ERA5), energy (SolarPower), communication (MobileNJ and MobileSH), and urban systems (CrowdBJ, CrowdBM, TaxiBJ, BikeDC and Los-Speed), each featuring distinct spatiotemporal characteristics. Detailed information on the datasets can be found in Appendix E.1.

**Baselines.** We compare against nine representative state-of-the-art baselines commonly adopted in spatiotemporal modeling, including: **Gaussian Process (GP)**, **DeepState** (Rangapuram et al., 2018), **D3VAE** (Li et al., 2022), **DiffSTG** (Wen et al., 2023), **TimeGrad** (Rasul et al., 2021), **CSDI** (Tashiro

Table 1: Short-term forecasting results in terms of CRPS, QICE, and IS. **Bold** indicates the best performance, while underlining denotes the second-best. DYffusion is limited to grid-format data, and "-" denotes results that are not applicable.

| Model | Climate | | | MobileSH | | | TaxiBJ | | | SolarPower | | | CrowdBJ | | |
|---|---|---|---|---|---|---|---|---|---|---|---|---|---|---|---|
| | CRPS | QICE | IS | CRPS | QICE | IS | CRPS | QICE | IS | CRPS | QICE | IS | CRPS | QICE | IS |
| GP | 0.083 | 0.158 | 9.98 | 0.495 | 0.120 | 6.90 | 0.217 | 0.137 | 258.9 | 0.732 | 0.222 | 769.0 | 0.601 | 0.152 | 17.1 |
| DeepState | 0.027 | 0.010 | 5.05 | 0.441 | 0.043 | 0.651 | 0.384 | 0.050 | 470.23 | 0.654 | 0.097 | 656.8 | 0.630 | 0.054 | 34.7 |
| D3VAE | 0.053 | 0.071 | 15.8 | 0.856 | 0.105 | 1.73 | 0.433 | 0.160 | 985.7 | 0.475 | 0.083 | 731.1 | 0.668 | 0.099 | 53.6 |
| DiffSTG | 0.026 | 0.068 | 7.42 | 0.303 | 0.078 | 0.526 | 0.299 | 0.074 | 416.5 | 0.213 | 0.068 | 240.6 | 0.436 | 0.089 | 32.1 |
| TimeGrad | 0.042 | 0.147 | 16.0 | 0.489 | 0.143 | 0.759 | 0.170 | 0.102 | 213.2 | 1.000 | 0.128 | 781.7 | 0.385 | 0.113 | 48.6 |
| CSDI | 0.027 | 0.019 | 5.18 | 0.200 | 0.052 | 0.295 | 0.122 | 0.048 | 121.8 | 0.267 | 0.050 | 221.6 | 0.306 | 0.028 | 16.4 |
| TMDM | 0.198 | 0.127 | 17.4 | 1.81 | 0.126 | 14.1 | 0.493 | 0.113 | 961.0 | 0.845 | 0.124 | 992.7 | 1.48 | 0.127 | 77.4 |
| NPDiff | 0.022 | 0.031 | 4.24 | 0.201 | 0.106 | 0.627 | 0.222 | 0.112 | 474.1 | 0.209 | 0.020 | **175.3** | 0.287 | 0.120 | 34.5 |
| DYffusion | **0.020** | 0.123 | 12.4 | 0.230 | 0.096 | 0.573 | **0.084** | 0.054 | 99.5 | - | - | - | - | - | - |
| **CoST** | 0.021 | **0.009** | **4.04** | **0.147** | **0.014** | **0.215** | 0.100 | **0.023** | **95.3** | 0.208 | **0.019** | 192.1 | **0.215** | **0.014** | **11.5** |

Table 2: Short-term forecasting results in terms of MAE and RMSE.

| Model | Climate | | MobileSH | | TaxiBJ | | SolarPower | | CrowdBJ | |
|---|---|---|---|---|---|---|---|---|---|---|
| | MAE | RMSE | MAE | RMSE | MAE | RMSE | MAE | RMSE | MAE | RMSE |
| GP | 1.51 | 1.64 | 0.359 | 0.611 | 28.0 | 30.5 | 43.9 | 97.1 | 3.37 | 4.70 |
| DeepState | 0.960 | 1.21 | 0.100 | 0.135 | 55.3 | 77.0 | 41.7 | 78.5 | 5.64 | 8.04 |
| D3VAE | 1.75 | 2.31 | 0.186 | 0.373 | 49.3 | 84.8 | 60.1 | 122.8 | 5.16 | 10.1 |
| DiffSTG | 0.90 | 1.13 | 0.066 | 0.103 | 41.8 | 69.4 | 31.1 | 63.8 | 3.68 | 6.63 |
| TimeGrad | 1.31 | 1.48 | 0.047 | 0.053 | 29.1 | 34.1 | 39.3 | 94.8 | 4.37 | 5.43 |
| CSDI | 0.94 | 1.20 | 0.044 | 0.075 | 18.2 | 21.6 | 38.8 | 69.6 | 2.71 | 4.50 |
| TMDM | 4.29 | 5.38 | 0.526 | 0.660 | 74.5 | 96.7 | 65.2 | 137.9 | 12.8 | 18.5 |
| NPDiff | 0.79 | 1.07 | 0.037 | 0.057 | 26.7 | 52.2 | 32.1 | 53.6 | 2.05 | 3.27 |
| DYffusion | 0.86 | 1.07 | 0.050 | 0.072 | **12.3** | **18.0** | - | - | - | - |
| **CoST** | **0.74** | **0.96** | **0.033** | **0.051** | 15.1 | 25.6 | **29.7** | **51.9** | **1.92** | **3.04** |

et al., 2021), **DYffusion** (Rühling Cachay et al., 2023), **TMDM** (Li et al., 2024), and **NPDiff** (Sheng et al., 2025). Detailed descriptions of each baseline are provided in Appendix E.2.

**Metrics.** We evaluate performance using two deterministic metrics (MAE, RMSE) and three probabilistic metrics (CRPS, QICE, IS). For QICE, we set $M_{\text{QIs}} = 10$ bins following its original design (Han et al., 2022), which offers a balanced trade-off between granularity and stability. For IS, we choose a confidence level of 90% (i.e., $\alpha_{CI} = 0.1$) following common practice in spatiotemporal forecasting tasks (Tashiro et al., 2021; Rasul et al., 2021).

**Experimental configuration.** We define short-term forecasting as predicting the next 12 steps from the previous 12 observations (Sheng et al., 2025; Wen et al., 2023), and long-term forecasting as predicting the next 64 steps from the previous 64 (Yuan et al., 2024; Jin et al., 2023b). As temporal granularity varies across datasets, the actual durations differ. Full configurations are in Appendix E.3.

## 5.1 Spatiotemporal Probabilistic Forecasting

**Short-term forecasting.** Table 1 reports probabilistic metrics, with additional results in Appendix Table 6. CoST consistently outperforms baselines, achieving average improvements of 17.4% in CRPS, 46.6% in QICE, and 16.5% in IS. This indicates superior distribution modeling and more reliable prediction intervals. While not always best on every single metric, CoST remains competitive across datasets. Deterministic results (Table 2, Appendix Table 7) show reductions of 7% in MAE and 6.1% in RMSE, confirming that integration with a strong conditional mean estimator enhances regular pattern capture. In contrast, TMDM underperforms other baselines, mainly because its time-series–oriented design limits generalization to spatiotemporal data, and its focus on long sequences weakens short-term prediction, as evidenced in Appendix Tables 8 and 9. Further experiments on the ETTh1 and ETTh2 time-series datasets show notable performance gains (Appendix Table 11).

**Long-term forecasting.** As shown in Appendix Table 8, CoST achieves substantial improvements in long-term forecasting under probabilistic metrics, with an improvement of 15.0% and 70.4% in terms of CRPS and QICE. Despite adopting a simple MLP architecture, CoST achieves higher overall accuracy than CSDI, a Transformer-based model tailored for capturing long-range dependencies. Furthermore, it provides significantly better training efficiency and inference speed, as detailed in Section 5.4. In addition, CoST performs well on deterministic metrics (Appendix Table 9), achieving an average reduction of 9.0% in MAE and 11.0% in RMSE compared to the best-performing baseline.

**Framework generalization.** To demonstrate the generality of CoST, we instantiate it with four representative spatiotemporal forecasting models: STID (Shao et al., 2022), STNorm (Deng et al.,

Table 3: Performance of different deterministic backbone models within the CoST framework. "Diffusion (w/o m)" denotes the results obtained using a single diffusion model.

| Model | Climate | | | | | MobileNJ | | | | | BikeDC | | | | |
|---|---|---|---|---|---|---|---|---|---|---|---|---|---|---|---|
| | MAE | RMSE | CRPS | QICE | IS | MAE | RMSE | CRPS | QICE | IS | MAE | RMSE | CRPS | QICE | IS |
| Diffusion (w/o m) | 1.070 | 1.361 | 0.030 | 0.030 | 6.58 | 0.195 | 0.6711 | 0.159 | 0.036 | 1.364 | 2.387 | 10.79 | 1.090 | 0.059 | 12.6 |
| +iTransformer | 0.818 | 1.088 | 0.023 | 0.018 | 4.83 | 0.122 | 0.207 | 0.123 | 0.021 | 0.815 | 0.526 | 2.23 | 0.454 | 0.035 | 3.82 |
| Reduction | 23.6% | 20.1% | 23.3% | 40.0% | 26.6% | 37.4% | 69.2% | 22.6% | 41.7% | 40.2% | 78.0% | 79.3% | 58.3% | 40.7% | 69.7% |
| + ConvLSTM | 0.889 | 1.151 | 0.027 | 0.024 | 5.54 | 0.137 | 0.231 | 0.120 | 0.025 | 0.913 | 0.454 | 2.01 | 0.443 | 0.037 | 6.07 |
| Reduction | 16.9% | 15.4% | 10.0% | 20.0% | 15.8% | 29.7% | 65.6% | 24.5% | 30.6% | 33.1% | 81.0% | 81.4% | 59.4% | 37.3% | 51.8% |
| +STNorm | 0.819 | 1.066 | 0.023 | 0.007 | 4.52 | 0.144 | 0.276 | 0.123 | 0.016 | 0.825 | 0.600 | 2.71 | 0.500 | 0.029 | 3.74 |
| Reduction | 23.5% | 21.7% | 23.3% | 76.7% | 31.3% | 26.2% | 58.9% | 22.6% | 55.6% | 39.5% | 74.9% | 74.9% | 54.1% | 50.8% | 70.3% |

2021), ConvLSTM (Shi et al., 2015), and iTransformer (Liu et al.). These models cover a diverse set of deep learning architectures, including CNNs, RNNs, MLPs, and Transformers. As shown in Table 3, CoST consistently enhances the performance of these backbones by effectively integrating deterministic and probabilistic modeling. Compared to using a single diffusion model, CoST yields more accurate predictions and better-calibrated uncertainty estimates, validating the framework's broad applicability and effectiveness.

**Case study of SST forecasting.** To assess our model's ability to quantify uncertainty under complex climate dynamics, we evaluate its performance in a key region for ENSO-related Sea Surface Temperature (SST) forecasting. As shown in Figure 3, our model produces high-fidelity SST forecasts that closely match ground truth across both warm pool and cold tongue regions. In addition to accurate mean predictions, it provides well-calibrated uncertainty estimates, revealing elevated variance in the central equatorial Pacific, especially near 0° latitude and 140°–130°W, where sharp thermocline gradients and non-linear feedbacks make forecasting particularly challenging. These high-uncertainty areas align with known regions of model divergence in climate science (McPhaden et al., 2006; Cane, 2005; Cao et al., 2021), demonstrating that our method delivers both accurate predictions and geophysically consistent uncertainty estimates.

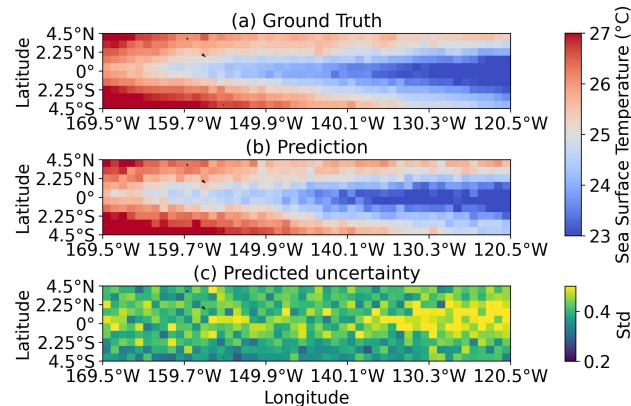

Figure 3: (a) and (b) show the ground-truth and predicted value of SST, and (c) displays the spatial distribution of forecasting uncertainty.

## 5.2 ABLATION STUDY

We perform an ablation study to assess the contribution of each proposed module. Specifically, we construct three model variants by progressively removing key components: **(w/o s)** removes the scale-aware diffusion process; **(w/o q)** excludes the customized fluctuation scale as a prior; **(w/o m)** removes the conditional mean predictor, relying solely on the diffusion model. Experiments on two datasets (Appendix Figure 8) show that the deterministic predictor notably improves performance by capturing regular spatiotemporal patterns, while also reducing the diffusion model's complexity. Adding the customized fluctuation scale further enhances accuracy, indicating its utility in providing valuable fluctuation information across different spatial units. And the scale-aware diffusion process enables the diffusion model to better utilize this condition.

## 5.3 QUALITATIVE ANALYSIS

**Analysis of distribution alignment.** As shown in Figure 4, the ground truth exhibits clear spatiotemporal multi-modality. In Figure 4(a), three peaks likely correspond to different time points or varying states at the same time. CoST accurately captures all three peaks, while CSDI only fits two, showing CoST's superior multi-modal modeling. In Figure 4(b), both models capture two peaks, but CoST aligns better with the peak spacing in the true distribution, reflecting stronger temporal

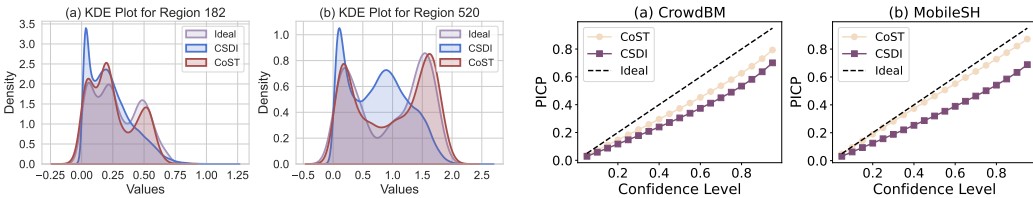

Figure 4: KDE plots of the MobileSH dataset for different regions: (a) Region 182, (b) Region 520.

Figure 5: PICP comparison between our model and CSDI on CrowdBM and MobileSH.

sensitivity. These strengths arise from CoST's hybrid design: the diffusion component models residual uncertainty to capture multi-modal traits, while the deterministic backbone learns regular trends. See Appendix E.5.1 for more analysis and results.

**Analysis of prediction quality.** To intuitively demonstrate the effectiveness of our predictions, we visualize results on the CrowdBJ dataset in Figure 6 (More Cases shown in Appendix Figure 10), comparing our model with the best baseline, CSDI. As shown in Figures 6 (a, c, f), our model, aided by a deterministic backbone, better captures regular spatiotemporal patterns. Meanwhile, the diffusion module enhances uncertainty modeling by focusing on residuals, as reflected in Figures 6 (b, d, e). Beyond sample-level comparison, we evaluate prediction interval calibration via dynamic quantile error curves on CrowdBM and MobileSH (Figure 5). For each confidence level $\alpha$, we compute the corresponding quantile interval and its Prediction Interval Coverage Probability (PICP). Closer alignment with the diagonal (black dashed line) indicates better calibration. Our model consistently outperforms CSDI in this regard.

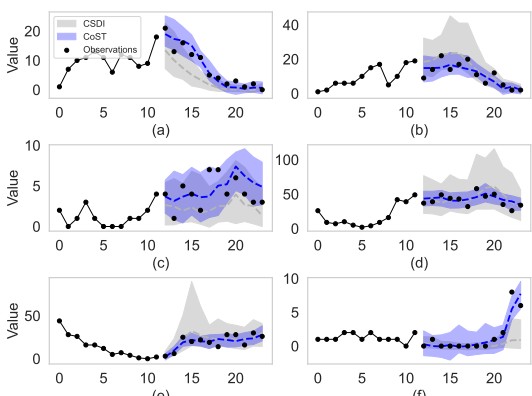

Figure 6: Visualizations of predictive uncertainty for both CSDI and CoST on the CrowdBJ dataset. The shaded regions represent the 90% confidence interval. The dashed lines denote the median of the predicted values for each model.

### 5.4 COMPUTATIONAL COST

We benchmark training and inference time (including 50 sampling iterations and pretraining for our mean predictor) on the MobileSH dataset. As shown in Appendix Table 10, CoST is markedly more efficient than existing probabilistic models. The efficiency comes from modeling only the residual distribution with a lightweight denoising network. In contrast, other baselines incur high computational costs by modeling the full data distribution with complex networks, while our approach is well-suited for time-sensitive applications such as mobile traffic prediction.

### 6 CONCLUSION

In this work, we highlight the importance of probabilistic forecasting for complex spatiotemporal systems and propose CoST, a collaborative framework that integrates deterministic and diffusion models. By decomposing data into a conditional mean and residuals, CoST bridges deterministic and probabilistic modeling, accurately capturing regular patterns and uncertainties across diverse spatiotemporal systems. Experiments on ten real-world datasets show that CoST outperforms state-of-the-art methods by 25% on average. Our approach offers an effective solution for combining precise pattern learning with uncertainty modeling in spatiotemporal forecasting.

**Limitations and future work.** CoST relies on a strong deterministic backbone, which may limit its applicability in domains lacking mature models. Moreover, it has not yet been validated on complex physical systems governed by PDEs or coupled dynamics. Future work will explore physics-informed extensions, adaptive decomposition, and more generalizable architectures.

## REPRODUCIBILITY STATEMENT

We provide detailed experimental settings in Section 5 and Appendix E. To facilitate the reproduction of our results, the source code and datasets used in this work have been made publicly available in an anonymous repository at: `https://anonymous.4open.science/r/CoST_17116`.

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

## A USAGE OF LLMS

In compliance with the stated policy, we report the use of a large language model (LLM) as a general-purpose assistance tool in the writing of this paper. Its application was confined to copy-editing and language polishing, such as correcting grammar and syntax. The LLM played no role in the research ideation or the generation of the substantive content of this work.

## B RELATED WORK

### B.1 SPATIOTEMPORAL DETERMINISTIC FORECASTING.

Deterministic forecasting of spatiotemporal systems focuses on point estimation. These models are typically trained with loss functions like MSE or MAE to learn the conditional mean $\mathbb{E}[y|x]$, capturing regular patterns. Common deep learning architectures include MLP-based (Shao et al., 2022; Qin et al., 2023; Zhang et al., 2023), CNN-based (Li et al., 2017; Liu et al., 2018; Zhang et al., 2017), and RNN-based (Bai et al., 2019b; Lin et al., 2020; Wang et al., 2017; 2018) models, valued for their efficiency. GNN-based methods (Bai et al., 2019a; 2020; Geng et al., 2019; Jin et al., 2023a) capture spatial dependencies in graph-based data, while Transformer-based models (Chen et al., 2022; 2021b; Ma et al., 2024; Yu et al., 2020; Boussif et al., 2023) are effective at modeling complex temporal dynamics.

### B.2 SPATIOTEMPORAL PROBABILISTIC FORECASTING.

The core of probabilistic forecasting lies in modeling uncertainty, aiming to capture the full data distribution (Yang et al., 2024; Tashiro et al., 2021). This is particularly suited for modeling the stochastic nature of spatiotemporal systems. While early methods include classical Bayesian approaches like Gaussian Processes (GP) (Roberts et al., 2013) and influential deep learning models such as DeepAR (Salinas et al., 2020) and DeepStateSpace (Rangapuram et al., 2018), recent advances have explored generative models such as GANs (Jin et al., 2022; Zhang et al., 2021), VAEs (Chen et al., 2021a; De Miguel et al., 2022), and diffusion models (Chai et al., 2024; Lin et al., 2024). Diffusion models, in particular, have gained traction for their ability to model complex distributions with stable training, yielding strong performance in spatiotemporal forecasting (Rasul et al., 2021; Rühling Cachay et al., 2023; Shen & Kwok, 2023; Sheng et al., 2025).

## C BACKGROUND

### C.1 GLOSSARY

We summarize all notations and symbols used throughout the paper in Table 4.

### C.2 SPATIOTEMPORAL DATA

Spatiotemporal data typically come in two forms: (i) **Grid-based data**, where the spatial dimension $V$ can be expressed in a two-dimensional form as $H \times W$, with $H$ and $W$ denoting height and width, respectively. (ii) **Graph-based data**, where $V$ denotes the number of nodes in a spatial graph $\mathcal{G} = (\mathcal{V}, \mathcal{E}, \mathcal{A})$, defined by its set of nodes $\mathcal{V}$, the set of edges $\mathcal{E}$ and the adjacency matrix $\mathcal{A}$. Its elements $a_{ij}$ show if there's an edge between node $i$ and $j$ in $\mathcal{V}$, $a_{ij} = 1$ when there's an edge and $a_{ij} = 0$ otherwise.

## D METHODOLOGY

### D.1 ALGORITHM

The training and inference procedures of CoST are summarized in Algorithm 1 and Algorithm 2, respectively.

Table 4: Glossary of notations and symbols used in this paper.

| Symbol | Used for |
|---|---|
| $\mathcal{G} = (\mathcal{V}, \mathcal{E}, \mathcal{A})$ | Graph structure where $\mathcal{V}$ is the node set, $\mathcal{E}$ is the edge set, and $\mathbf{A}$ is the adjacency matrix. |
| $\mathbf{x} \in \mathbb{R}^{T \times V \times C}$ | Spatiotemporal data. |
| $T$ | The length of spatiotemporal series. |
| $V$ | The number of spatial units. |
| $C$ | The number of feature dimensions. |
| $B$ | Batch size. |
| $P$ | Prediction horizon. |
| $M$ | Historical horizon. |
| $N$ | The number of diffusion steps. |
| $H$ | Height of the grid-based data. |
| $W$ | Width of the grid-based data. |
| $\mathbf{Q}$ | Customized fluctuation scale. |
| $\mathcal{M}$ | The variance tensor. |
| $\mathbf{S}$ | The random sign tensor. |
| $\{\cdot\}^{co}$ | Historical (conditional) term. |
| $\{\cdot\}^{ta}$ | Predicted (target) term. |
| $\{\cdot\}_n$ | Noisy data at $n$-th diffusion step. |
| $\mu$ | Mean. |
| $\mathbf{r}$ | Residual. |
| $\epsilon$ | Gaussian noise. |
| $\mathcal{K}$ | $K$ The set of indices for the selected FFT components. |
| $\{\beta_n\}_{n=1}^N$ | The noise schedule in the diffusion process. |
| $\alpha_n, \bar{\alpha}_n$ | $\alpha_n = 1 - \beta_n$, $\bar{\alpha}_n = \prod_{i=1}^n \alpha_i$. |
| $\epsilon_\theta(\cdot)$ | The denoising network with parameter $\theta$. |
| $\alpha_{CI}$ | Significance level for the prediction interval. |
| $\mathbb{1}_{(\cdot)}$ | Indicator function, which takes the value 1 when a certain condition is true, and 0 when the condition is false. |

---

**Algorithm 1** Training

---

1: **Stage 1: Pretraining of Deterministic Model** $\mathbb{E}_\theta$
2: **repeat**
3:     Estimate the conditional mean $\mathbb{E}_\theta[\mathbf{x}_0^{ta}|\mathbf{x}_0^{co}]$.
4:     Update $\mathbb{E}_\theta$ using the following loss function:

$$\mathcal{L}_2 = \left\| \mathbb{E}_\theta[\mathbf{x}_0^{ta}|\mathbf{x}_0^{co}] - \mathbf{x}_0^{ta} \right\|_2^2$$

5: **until** The model has converged.
6: **Stage 2: Training of Diffusion Model** $\epsilon_\theta$
7: **repeat**
8:     Initialize $n \sim \text{Uniform}(1, \ldots, N)$ and $\epsilon \sim \mathcal{N}(0, I)$.
9:     Calculate the target $\mathbf{r}_0^{ta} = \mathbf{x}_0^{ta} - \mathbb{E}_\theta[\mathbf{x}_0^{ta}|\mathbf{x}_0^{co}]$.
10:     Calculate noisy targets $\mathbf{r}_n^{ta}$ using Eq. (13).
11:     Update $\epsilon_\theta$ using the following loss function:

$$\mathcal{L}(\theta) = \left\| \epsilon - \epsilon_\theta(\mathbf{r}_n^{ta}, n|\mathbf{x}_0^{co}) \right\|_2^2$$

12: **until** The model has converged.

---

# E EXPERIMENTS

## E.1 DATASETS

In our experiments, we evaluate the proposed method on ten real-world datasets across four domains: **climate**, **energy**, **communication systems**, and **urban systems**. For climate forecasting, we train our models on the simulated SST-CESM2 dataset and evaluate them on the observational SST-ERA5 dataset, using the first 30 years for validation and the remaining years for testing. The remaining datasets are partitioned into training, validation, and test sets with a 6:2:2 ratio, and all datasets are standardized during training. Table 5 provides a summary of the datasets. The details are as follows:

- **Climate.** We utilize two datasets for sea surface temperature (SST) prediction in the Niño 3.4 region (5°S–5°N, 170°W–120°W), which is widely used for monitoring El Niño events: (i) SST-CESM2, simulated SST data from the CESM2-FV2 model of the CMIP6 project, covering the period from

---

**Algorithm 2** Inference

1: **Input:** Context data $\mathbf{x}_0^{co}$, customized fluctuation scale $\mathbf{Q}$, trained diffusion model $\epsilon_\theta$, trained deterministic model $\mathbb{E}_\theta$
2: **Output:** Target data $\mathbf{x}_0^{ta}$
3: Estimate the conditional mean $\mathbb{E}_\theta[\mathbf{x}_0^{ta}|\mathbf{x}_0^{co}]$
4: Sample $\mathbf{r}_N^{ta}$ from $\epsilon \sim \mathcal{N}(\mathbf{Q}, I)$
5: **for** $n = N$ to 1 **do**
6:     Estimate the noise $\epsilon_\theta(\mathbf{r}_n^{ta}, n|\mathbf{x}_0^{co})$
7:     Calculate the $\mu_\theta(\mathbf{r}_n^{ta}, n|\mathbf{x}_0^{co})$ using Eq. (14)
8:     Sample $\mathbf{r}_{n-1}^{ta}$ using Eq. (2)
9: **end for**
10: **Return:** $\mathbf{x}_0^{ta} = \mathbb{E}_\theta[\mathbf{x}_0^{ta}|\mathbf{x}_0^{co}] + \mathbf{r}_0^{ta}$

---

Table 5: The basic information of spatio-temporal data.

| Dataset | Location | Type | Temporal Period | Spatial partition | Interval |
|---|---|---|---|---|---|
| SST-CESM2 | Global (Niño 3.4) | Simulated SST | 1850-2014 | $1° \times 1°$ | Monthly |
| SST-ERA5 | Global (Niño 3.4) | Reanalysis SST / U10 / V10 | 1940-2025 | $0.25° \times 0.25°$ | Monthly |
| SolarPower | China (a PV station) | GHI / Weather / PV power | 2024/03/01 - 2024/12/31 | Station-level | 15 min |
| TaxiBJ | Beijing | Taxi flow | 2014/03/01 - 2014/06/30 | $32 \times 32$ | Half an hour |
| BikeDC | Washington, D.C. | Bike flow | 2010/09/20 - 2010/10/20 | $20 \times 20$ | Half an hour |
| MobileSH | Shanghai | Mobile traffic | 2014/08/01 - 2014/08/21 | $32 \times 28$ | One hour |
| MobileNJ | Nanjing | Mobile traffic | 2021/02/02 - 2021/02/22 | $20 \times 28$ | One hour |
| CrowdBJ | Beijing | Crowd flow | 2018/01/01 - 2018/01/31 | 1010 | One hour |
| CrowdBM | Baltimore | Crowd flow | 2019/01/01 - 2019/05/31 | 403 | One hour |
| Los-Speed | Los Angeles | Traffic speed | 2012/03/01 - 2012/03/07 | 207 | 5 min |

1850 to 2014, with a spatial resolution of $1° \times 1°$. (ii) SST-ERA5: reanalysis data from ERA5, containing SST and 10-meter wind speed (U10/V10) variables from 1940 to 2025, with an original spatial resolution of approximately $0.25° \times 0.25°$. All data are regridded to a $1° \times 1°$ resolution for consistency. The CESM2 data are used for training, while the first 30 years of ERA5 are used for validation and the remaining years for testing.

- **Energy.** This dataset contains real-time meteorological measurements and photovoltaic (PV) power output collected from a PV power station in China, spanning from March 1st to December 31st, 2024. The features include: total active power output of the PV grid-connection point (P), ambient temperature, back panel temperature, dew point, relative humidity, atmospheric pressure, global horizontal irradiance (GHI), diffuse and direct radiation, wind direction and wind speed. Our forecasting task focuses on GHI, which is the key variable for solar power prediction. Due to data privacy restrictions, the raw dataset cannot be publicly released.

- **Communication Systems.** Mobile communication traffic datasets are collected from two major cities in Shanghai and Nanjing, capturing the spatiotemporal dynamics of network usage patterns.

- **Urban Systems.** We adopt five widely used public datasets representing various urban sensing signals: (i) CrowdBJ and CrowdBM, crowd flow data from Beijing and Baltimore, respectively. (ii) TaxiBJ, taxi trajectory-based traffic flow data from Beijing. (iii) BikeDC, bike-sharing demand data from Washington D.C. (iv) Los-Speed, traffic speed data from the Los Angeles road network. These datasets have been extensively used in spatiotemporal forecasting research and provide diverse signals for evaluating model generality across cities and domains.

### E.2 BASELINES

We provide a brief description of the baselines used in our experiments:

- **GP (Gaussian Processes):** A non-parametric time series forecasting method that models data as a Gaussian process, offering uncertainty estimates and effective modeling of non-linear relationships.

- **DeepState Rangapuram et al. (2018):** A deep learning framework for time series forecasting that integrates state space models with neural networks, enabling efficient probabilistic predictions by learning latent states and observation processes.

- **D3VAE (Li et al., 2022):** Aims at short-period and noisy time series forecasting. It combines generative modeling with a bidirectional variational auto-encoder, integrating diffusion, denoising, and disentanglement.

- **DiffSTG (Wen et al., 2023):** First applies diffusion models to spatiotemporal graph forecasting. By combining STGNNs and diffusion models, it reduces prediction errors and improves uncertainty modeling.

- **TimeGrad (Rasul et al., 2021):** An autoregressive model based on diffusion models. It conducts probabilistic forecasting for multivariate time series and performs well on real-world datasets.

- **CSDI (Tashiro et al., 2021):** Utilizes score-based diffusion models for time series imputation. It can leverage the correlations of observed values and also shows remarkable results on prediction tasks.

- **DYffusion (Rühling Cachay et al., 2023):** A training method for diffusion models in probabilistic spatiotemporal forecasting. It combines data temporal dynamics with diffusion steps and performs well in complex dynamics forecasting.

- **TMDM (Li et al., 2024):** TMDM integrates transformers with diffusion models for probabilistic time series forecasting, using transformer-based prior knowledge to enable accurate distribution forecasting and uncertainty estimation.

- **NPDiff (Sheng et al., 2025):** A general noise prior framework for mobile traffic prediction. It uses the data dynamics to calculate noise priors for the denoising process and achieve effective performance.

### E.3 EXPERIMENTAL CONFIGURATION

In our experiment, for our model, we set the training maximum epoch for both the deterministic model and the diffusion model to 50, with early stopping based on a patience of 5 for both models. For the diffusion model, we set the validation set sampling number to 3, and the average metric computed over these samples is used as the criterion for early stopping. For the baseline models, we set the maximum training epoch to 100 and the early stopping patience also to 5. We set the number of samples to 50 for computing the experimental results presented in the paper. For the denoising network architecture, we adopt a lightweight variant of the MLP-based STID (Shao et al., 2022). Specifically, we set the number of encoder layers to 8 and the embedding dimension to 128. The diffusion model employs a maximum of 50 diffusion steps, using a linear noise schedule with $\beta_1 = 0.0001$ and $\beta_N = 0.5$. During training, we set the initial learning rate to 0.001, and after 20 epochs, we adjust it to 4e-4. We use the Adam optimizer with a weight decay of 1e-6. All experiments are conducted with fixed random seeds. Models with lower GPU memory demands are run on NVIDIA TITAN Xp (12GB GDDR5X) and NVIDIA GeForce RTX 4090 (24GB GDDR6X) GPUs under a Linux environment. For the DYffusion (Rühling Cachay et al., 2023) baseline, which requires substantially more resources, training is performed on NVIDIA A100 (80GB HBM2e) and A800 (40GB HBM2e).

### E.4 GEOGRAPHIC EXTENT OF THE ENSO REGION

To provide geographic context for the SST case study presented in Section 3, Figure 7 illustrates the global location and spatial extent of the selected region. The red box highlights the area from 4.5°S to 4.5°N and 169.5°W to 120.5°W in the central-to-eastern equatorial Pacific, a region known for strong ocean-atmosphere coupling and ENSO-related variability.

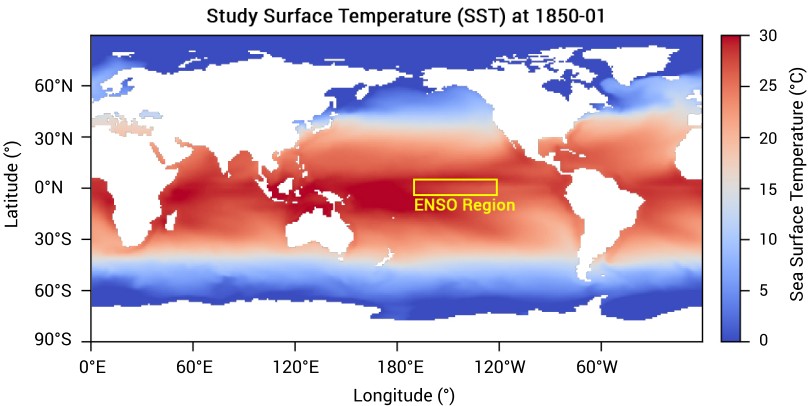

Figure 7: Global map indicating the spatial extent of ENSO region (highlighted in yellow). The region spans from 4.5°S to 4.5°N and 169.5°W to 120.5°W in the equatorial Pacific.

Table 6: Short-term forecasting results in terms of CRPS, QICE, and IS. **Bold** indicates the best performance, while underlining denotes the second-best. DYffusion is limited to grid-format data, and "-" denotes results that are not applicable.

| Model | BikeDC | | | MobileNJ | | | CrowdBM | | | Los-Speed | | |
|---|---|---|---|---|---|---|---|---|---|---|---|---|
| | CRPS | QICE | IS | CRPS | QICE | IS | CRPS | QICE | IS | CRPS | QICE | IS |
| GP | 0.494 | 0.120 | 6.69 | 0.435 | 0.129 | 4.76 | 0.620 | 0.159 | 66.8 | 0.789 | 0.1579 | 61.7 |
| DeepState | 0.728 | 0.084 | 15.5 | 0.518 | 0.065 | 4.40 | 0.689 | 0.057 | 97.0 | 0.086 | 0.040 | 40.9 |
| D3VAE | 0.785 | 0.157 | 8.77 | 0.565 | 0.096 | 6.03 | 0.593 | 0.110 | 136.4 | 0.119 | 0.089 | 90.5 |
| DiffSTG | 0.692 | 0.157 | 8.08 | 0.291 | 0.071 | 3.11 | 0.453 | 0.047 | 68.5 | 0.078 | 0.045 | 50.9 |
| TimeGrad | 0.469 | 0.130 | 5.65 | 0.432 | 0.162 | 5.87 | **0.240** | 0.085 | 46.9 | **0.031** | 0.098 | **20.8** |
| CSDI | 0.529 | 0.057 | 4.79 | 0.111 | 0.039 | 0.80 | 0.390 | 0.054 | 61.1 | 0.059 | 0.026 | 30.8 |
| TMDM | 2.32 | 0.125 | 29.6 | 1.49 | 0.126 | 87.5 | 3.46 | 0.124 | 217.3 | 0.897 | 0.126 | 83.4 |
| NPDiff | 0.442 | 0.066 | 7.11 | 0.128 | 0.133 | 2.22 | 0.331 | 0.119 | 91.2 | 0.057 | 0.023 | 30.5 |
| DYffusion | 0.573 | 0.079 | 6.46 | 0.196 | 0.080 | 1.80 | - | - | - | - | - | - |
| **CoST** | **0.419** | **0.028** | **3.45** | **0.089** | **0.032** | **0.66** | 0.256 | **0.027** | **37.8** | 0.056 | **0.023** | 31.9 |

## E.5 ADDITIONAL EXPERIMENTAL RESULTS

Table 7: Short-term forecasting results in terms of MAE and RMSE. **Bold** indicates the best performance, while underlining denotes the second-best. DYffusion is limited to grid-format data, and "-" denotes results that are not applicable.

| Model | BikeDC | | MobileNJ | | CrowdBM | | Los-Speed | |
|---|---|---|---|---|---|---|---|---|
| | MAE | RMSE | MAE | RMSE | MAE | RMSE | MAE | RMSE |
| GP | 0.941 | 1.74 | 0.257 | 0.682 | 6.35 | 17.7 | 6.60 | 11.0 |
| DeepState | 1.98 | 3.81 | 0.582 | 0.827 | 13.9 | 23.2 | 6.50 | 9.23 |
| D3VAE | 0.871 | 3.59 | 0.580 | 1.135 | 11.0 | 24.7 | 8.28 | 11.9 |
| DiffSTG | 0.770 | 4.02 | 0.317 | 0.649 | 8.88 | 21.3 | 5.38 | 9.75 |
| TimeGrad | 0.843 | **1.07** | 0.340 | 0.357 | 10.1 | 12.4 | **2.33** | **3.00** |
| CSDI | 0.592 | 3.10 | 0.129 | 0.237 | 7.31 | 19.3 | 4.53 | 8.07 |
| TMDM | 2.44 | 4.11 | 3.27 | 4.10 | 72.9 | 94.8 | 9.42 | 13.9 |
| NPDiff | **0.435** | 1.90 | 0.123 | 0.175 | 5.42 | 13.7 | 4.07 | 7.64 |
| DYffusion | 0.480 | 1.37 | 0.222 | 0.357 | - | - | - | - |
| CoST | 0.492 | 1.76 | **0.102** | **0.172** | **5.04** | **12.1** | 4.05 | 7.30 |

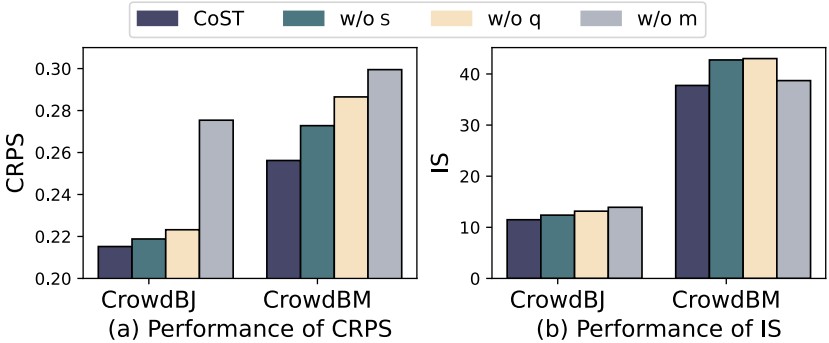

Figure 8: Ablation study on the CrowdBJ and CrowdBM comparing variants in terms of (a) CRPS and (b) IS.

Table 10: Comparison of training and inference time on the MobileSH dataset.

| Model | Train Time | Inference Time |
|---|---|---|
| D3VAE | 3min 27s | 2min 15s |
| DiffSTG | 24min 16s | 18min 38s |
| TimeGrad | 5min | 2min |
| CSDI | 48min 40s | 38min 49s |
| DyDiffusion | 33h | 3h |
| CoST | 2min | 50s |

Table 8: Long-term forecasting results in terms of CRPS, QICE, and IS. **Bold** indicates the best performance, while underlining denotes the second-best. DYffusion is limited to grid-format data, and "-" denotes results that are not applicable.

| Model | MobileSH | | | Climate | | | CrowdBJ | | | CrowdBM | | | Los-Speed | | |
|---|---|---|---|---|---|---|---|---|---|---|---|---|---|---|---|
| | CRPS | QICE | IS | CRPS | QICE | IS | CRPS | QICE | IS | CRPS | QICE | IS | CRPS | QICE | IS |
| GP | 0.537 | 0.112 | 7.13 | 0.086 | 0.146 | 9.18 | 0.660 | 0.143 | 19.4 | 0.622 | 0.153 | 76.5 | 0.910 | 0.149 | 112.6 |
| DeepState | 0.707 | 0.066 | 0.924 | 0.031 | 0.018 | 6.09 | 0.925 | 0.073 | 42.4 | 1.02 | 0.080 | 123.2 | 0.133 | 0.090 | 68.5 |
| D3VAE | 0.798 | 0.129 | 1.830 | 0.075 | 0.083 | 24.0 | 0.710 | 0.109 | 63.9 | 0.674 | 0.108 | 152.3 | 0.138 | 0.101 | 113.2 |
| DiffSTG | 0.374 | 0.107 | 0.923 | 0.027 | 0.077 | 7.90 | 0.370 | 0.094 | 31.3 | 0.400 | 0.073 | 67.1 | 0.124 | 0.080 | 104.6 |
| TimeGrad | 0.245 | 0.075 | 0.408 | 0.041 | 0.101 | 14.2 | 0.371 | 0.073 | 32.4 | 0.237 | 0.049 | 33.9 | 0.192 | 0.081 | 98.8 |
| CSDI | 0.158 | 0.045 | **0.216** | 0.036 | 0.073 | 6.80 | 0.229 | 0.038 | 12.0 | 0.235 | 0.052 | 33.7 | 0.134 | 0.090 | **59.2** |
| TMDM | 0.799 | 0.127 | 16.1 | 0.093 | 0.115 | 7.36 | 0.751 | 0.127 | 77.5 | 0.346 | 0.125 | 187.7 | 0.904 | 0.121 | 837.0 |
| NPDiff | 0.204 | 0.102 | 0.611 | 0.109 | 0.115 | 41.3 | 0.288 | 0.114 | 33.6 | 0.331 | 0.111 | 90.8 | 1.366 | 0.126 | 950.4 |
| DYffusion | 0.308 | 0.086 | 0.550 | 0.030 | 0.147 | 15.2 | - | - | - | - | - | - | - | - | - |
| **CoST** | **0.158** | **0.016** | 0.218 | **0.024** | **0.011** | **4.87** | **0.217** | **0.011** | **11.5** | **0.235** | **0.009** | **31.2** | **0.089** | **0.040** | 64.6 |

Table 9: Long-term forecasting results in terms of MAE and RMSE. **Bold** indicates the best performance, while underlining denotes the second-best. DYffusion is limited to grid-format data, and "-" denotes results that are not applicable.

| Model | MobileSH | | SST | | CrowdBJ | | CrowdBM | | Los-Speed | |
|---|---|---|---|---|---|---|---|---|---|---|
| | MAE | RMSE | MAE | RMSE | MAE | RMSE | MAE | RMSE | MAE | RMSE |
| GP | 0.399 | 0.627 | 1.52 | 1.65 | 2.52 | 3.75 | 7.37 | 20.7 | 6.69 | 11.2 |
| DeepState | 0.160 | 0.199 | 1.13 | 1.40 | 8.53 | 11.0 | 21.5 | 31.9 | 10.1 | 14.2 |
| D3VAE | 0.207 | 0.392 | 2.39 | 3.13 | 5.63 | 11.4 | 12.4 | 28.2 | 9.43 | 13.3 |
| DiffSTG | 0.078 | 0.125 | 0.94 | 1.19 | 3.04 | 6.37 | 7.59 | 18.8 | 7.77 | 14.2 |
| TimeGrad | 0.058 | 0.072 | 1.30 | 1.64 | 3.48 | 4.83 | 5.25 | **7.40** | 18.2 | 22.3 |
| CSDI | **0.035** | 0.057 | 1.31 | 1.63 | 1.99 | 3.64 | **4.64** | 12.4 | 11.3 | 15.0 |
| TMDM | 0.519 | 6.50 | 1.55 | 1.73 | 3.54 | 8.32 | 15.2 | 29.0 | 34.2 | 43.1 |
| NPDiff | 0.037 | 0.057 | 1.91 | 2.82 | 2.06 | 3.28 | 5.44 | 13.8 | 46.0 | 58.3 |
| DYffusion | 0.047 | 0.066 | **0.85** | **1.06** | - | - | - | - | - | - |
| **CoST** | 0.035 | **0.053** | 0.86 | 1.13 | **1.92** | **3.05** | 4.74 | 11.2 | **5.94** | **10.8** |

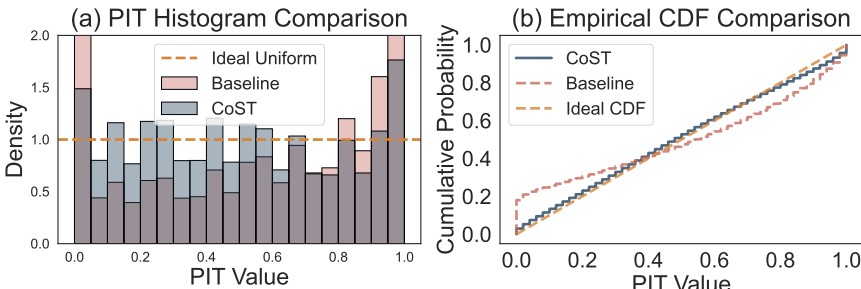

Figure 9: PIT analysis on the MobileSH dataset: (a) PIT histogram and (b) PIT empirical CDF.

Table 11: Long-term forecasting performance comparison of TMDM on ETTh1 and ETTh2 Datasets.

| Model | EETh1 | | | EETh2 | | |
|-------|-------|------|-----|-------|------|-----|
| | CRPS | QICE | IS | CRPS | QICE | IS |
| TMDM | 0.395 | 0.041 | 4.8 | 0.196 | 0.018 | 2.2 |
| **CoST** | 0.311 | 0.007 | 1.6 | 0.109 | 0.007 | 0.78 |

### E.5.1 ANALYSIS OF DISTRIBUTION ALIGNMENT.

Additionally, we present the PIT (Probability Integral Transform) histogram in Figure 9 (a) and the PIT empirical cumulative distribution function (CDF) in Figure 9 (b) to visually reflect the alignment of the full distribution. Ideally, the true values' quantiles in the predictive distribution should follow a uniform distribution, corresponding to the dashed line in Figure 9 (a). In the case of perfect calibration, the PIT CDF should closely resemble the yellow diagonal line. Clearly, our model outperforms CSDI.

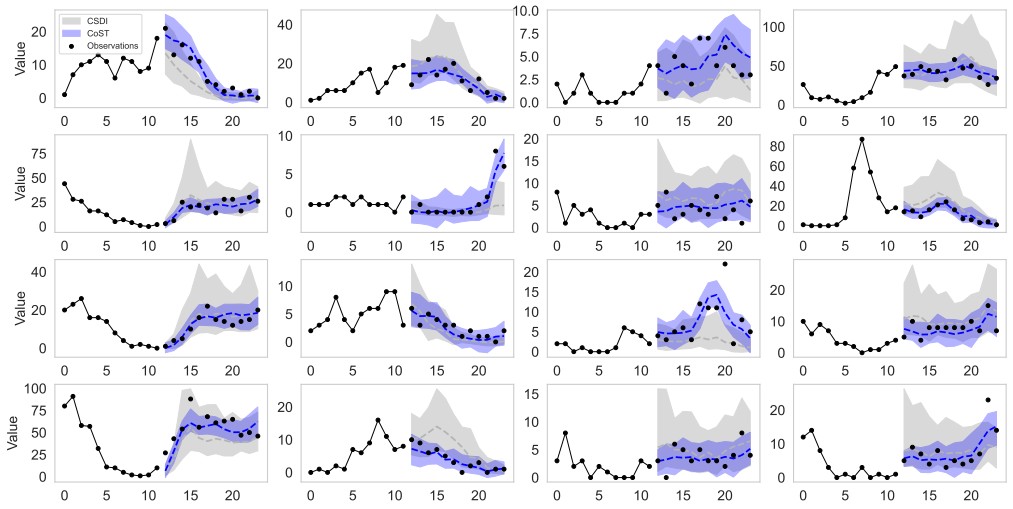

Figure 10: Visualizations of predictive uncertainty for both CSDI and CoST on the CrowdBJ dataset. The shaded regions represent the 90% confidence interval. The dashed lines denote the median of the predicted values for each model.

