# OpenReview forum: "Collaborative Deterministic–Probabilistic Forecasting for Diverse Spatiotemporal Systems"
_ICLR.cc/2026/Conference — Submitted to ICLR 2026_

### Official Review · Reviewer_itWV · 2025-10-18

**Soundness:** 2
**Presentation:** 3
**Contribution:** 1
**Rating:** 4
**Confidence:** 3

**Summary:**

The paper proposes CoST, a hybrid framework that combines deterministic and diffusion-based probabilistic forecasting for spatiotemporal systems. Using a mean–residual decomposition and scale-aware diffusion, CoST improves both accuracy and uncertainty estimation, achieving state-of-the-art performance with lower computational cost across multiple real-world datasets.

The authors use several new datasets as backbones and compare their method with recent approaches, but I noticed that some of these datasets are not open-sourced. It is recommended to include results on publicly available datasets.

The paper should also provide more qualitative comparisons, especially with published works, to show how the predicted distributions differ.

The authors need to clarify how their motivation differs from that of TMDM and what the core contribution of this paper truly is.

Apart from introducing FFT, what are the other architectural differences from TMDM?

Since QICE has already been widely applied in probabilistic time series forecasting, it does not need to occupy so much space in the paper.

**Strengths:**

see summary

**Weaknesses:**

see summary

**Questions:**

see summary

---

### Official Review · Reviewer_HJmL · 2025-10-28

**Soundness:** 3
**Presentation:** 3
**Contribution:** 2
**Rating:** 4
**Confidence:** 4

**Summary:**

The paper proposes **CoST**, a collaborative framework that first learns the **conditional mean** with a strong deterministic backbone and then models the **residual** with a lightweight diffusion model.

**Strengths:**

- **S1** The overall design is solid and well-structured. The “deterministic mean + diffusion residual” idea is clearly laid out with intuitive motivation and a concrete algorithmic implementation.
- **S2** Broad experimental coverage (ten datasets) with improvements in both probabilistic and point metrics; the paper also claims efficiency benefits.

**Weaknesses:**

- **W1** One of the claimed core contributions—the Mean-Residual Decomposition Framework—appears to be highly similar to the approach proposed by Li et al. (ICLR 2025) [1].A lthough the authors emphasize that their work targets spatiotemporal data, the design of combining a deterministic model with a diffusion model does not exhibit substantial differences from the prior work. Hence, the novelty of this contribution is questionable.

- **W2** The paper lacks quantitative evidence to support its analysis of the characteristics of deterministic and uncertain components (such as residuals) in spatiotemporal data.

- **W3** In the residual component construction stage (Eq. 9), components with magnitudes below $0.1×A_{max}$ are treated as residuals. The constant 0.1 should be considered a hyperparameter; however, the paper does not include a sensitivity analysis regarding this parameter. This omission is important because the choice of this threshold may affect the model’s transferability across different data domains.

[1]. Li, Q., Zhang, Z., Yao, L., Li, Z., Zhong, T., & Zhang, Y. (2025). Diffusion-based decoupled deterministic and uncertain framework for probabilistic multivariate time series forecasting. In The Thirteenth International Conference on Learning Representations.

**Questions:**

- **Q1** From the motivation perspective, this work appears to be an extension of the decoupling deterministic and uncertain components idea in D3U [1], moving from temporal systems to spatiotemporal systems. Could the authors further elaborate on the key distinctions between CoST and D3U in this regard?

- **Q2** It is recommended that the authors include a sensitivity analysis for the hyperparameter mentioned in Weakness-3 and provide empirical guidance on how to set this parameter effectively.

[1]. Li, Q., Zhang, Z., Yao, L., Li, Z., Zhong, T., & Zhang, Y. (2025). Diffusion-based decoupled deterministic and uncertain framework for probabilistic multivariate time series forecasting. In The Thirteenth International Conference on Learning Representations.

---

### Official Review · Reviewer_Z8Ba · 2025-10-31

**Soundness:** 2
**Presentation:** 2
**Contribution:** 2
**Rating:** 4
**Confidence:** 3

**Summary:**

This paper proposes a forecasting framework called CoST, which integrates deterministic and diffusion models to handle diverse spatiotemporal systems. CoST employs a mean–residual decomposition strategy, where a deterministic model predicts the conditional mean, and a diffusion model captures the residual uncertainties. Extensive experiments demonstrate that CoST outperforms state-of-the-art baselines.

**Strengths:**

1. The authors provide a solid theoretical proof supporting the rationality of the proposed mean–residual decomposition strategy.

2.  Extensive experiments convincingly demonstrate the effectiveness of the proposed model.

3. The figures are clearly presented and easy to understand.

**Weaknesses:**

1. The authors identify three challenges in the introduction. However, the introduction does not clearly explain how CoST addresses each of them individually.
2. Equations 10 and 11 lack sufficient explanation of their underlying motivation. Moreover, the parameters $p$ and $P$ are not explicitly defined.
3. Although this paper focuses on spatiotemporal prediction tasks, CoST does not explicitly demonstrate how it captures spatial correlations. Furthermore, the ablation study omits a variant that enforces identical variance across all locations.
4. Since pre-trained existing models are used to predict the mean, evaluating prediction performance using MAE and RMSE is not appropriate. This setup is equivalent to directly using the outputs of the existing models, which prevents a fair assessment of CoST's effectiveness.
5. The paper does not specify which deterministic model CoST employs in Tables 1 and 2.

**Questions:**

See Weaknesses 1–4 for details. In addition, many existing methods also employ a similar mean–residual decomposition strategy. What are the fundamental differences between CoST and these existing approaches?

---

### Meta-Review · Area_Chair_sRJj · 2026-01-05

**Summary:**

Reviewers consistently question the clarity and novelty of the core contributions, noting insufficient explanation of key design choices and weak differentiation from closely related work. The experimental evaluation is also considered inadequate, with concerns about metric suitability, missing ablations, limited analysis, and lack of results on public datasets. Moreover, the authors did not provide any response during the rebuttal stage. Considering these issues, I recommend rejection.

**Reviewer Concerns:**

All reviews are still outstanding.

**Reviewer Scores:**

There will be no change.

---

### Decision · Program_Chairs · 2026-01-26

Reject